# NEURREV: TRAIN BETTER SPARSE NEURAL NETWORK PRACTICALLY VIA NEURON REVITALIZATION

**Gen Li**[1], **Lu Yin**[2,5], **Jie Ji**[1], **Wei Niu**[3], **Minghai Qin**, **Bin Ren**[4], **Linke Guo**[1],
**Shiwei Liu**[2,6], **Xiaolong Ma**[1]

[1] Clemson University [2] Eindhoven University of Technology [3] University of Georgia
[4] William & Mary [5] University of Aberdeen [6] University of Oxford
`{gen,xiaolom}@clemson.edu`

## ABSTRACT

Dynamic Sparse Training (DST) employs a greedy search mechanism to identify an optimal sparse subnetwork by periodically pruning and growing network connections during training. To guarantee effectiveness, DST algorithms rely on high search frequency, which consequently, requires large learning rate and batch size to enforce stable neuron learning. Such settings demand extreme memory consumption, as well as generating significant system overheads that limit the wide deployment of deep learning-based applications on resource-constraint platforms. To reconcile such, we propose an Neuron Revitalizationframework for DST (NeurRev), based on an innovative finding that dormant neurons exist in the presence of weight sparsity and cannot be revitalized (i.e., activated for learning) even with a high sparse mask search frequency. These dormant neurons produce a large quantity of zeros during training, which contribute relatively little to the outputs of succeeding layers or to the final results. Different from most existing DST algorithms that spare no effort designing weight-growing criteria, NeurRev focuses on optimizing the long-neglected pruning part, which awakens dormant neurons by pruning and incurs no additional computation costs. As such, NerRev advances more effective neuron learning, which not only achieves outperformance accuracy in a variety of networks and datasets but also promotes low-cost dynamism at the system level. Systematical evaluations on training speed and system overhead are conducted on mobile devices, where the proposed NeurRev framework consistently outperforms representative state-of-the-arts. Code available in `https://github.com/coulsonlee/NeurRev-ICLR2024`.

## 1 INTRODUCTION

Deep Neural Networks (DNNs) have gained significant traction in a number of industries, such as advanced manufacturing (Nguyen et al., 2021; Zhang et al., 2021; Liu et al., 2020), connected healthcare (Nguyen et al., 2022; 2021), and Augmented Reality/Virtual Reality (AR/VR) (Chen et al., 2020; Eriş et al., 2021), etc. As the technologies promote a plethora of emerging applications, edge computing (Shi et al., 2016; Abbas et al., 2017) becomes the core enabler to facilitate the widespread of machine intelligence. However, as large models are prevailing in the recent deep learning era, matching the resource budget and the ever increasing DNN model complexity becomes unprecedented challenging (Li et al., 2019; Liu et al., 2022a; Li et al., 2023).

With the potential to save time in both training and inference, dynamic sparse training (DST) (Mocanu et al., 2018; Mostafa & Wang, 2019; Dettmers & Zettlemoyer, 2019; Lym et al., 2019; Evci et al., 2020; Jayakumar et al., 2020; Yuan et al., 2021; Liu et al., 2021c;a; Hou et al., 2022; Jaiswal et al., 2022; Yin et al., 2024) has received tremendous attention for extending the scalability and facilitating larger design space of deep intelligence implementations (Liu et al., 2021b; Yuan et al., 2022). Due to its "always sparse" training regime and impressive training accuracy, DST maintains low memory footprint by avoiding the full dense model computation, and quickly becomes a preeminent approach in efficient training field.

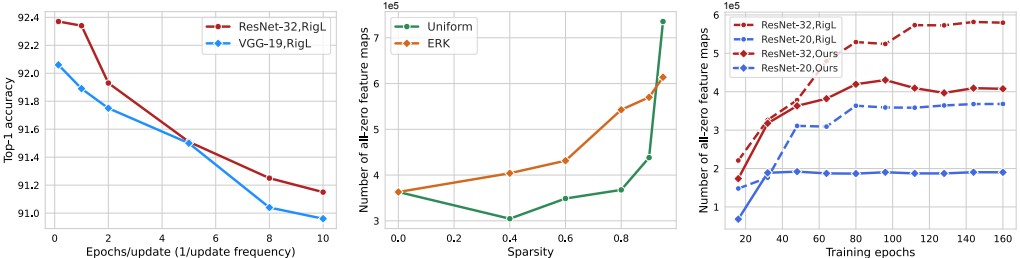

Figure 1: Preliminary results of DST. Left: accuracy changes with different DST update frequency. Middle: dormant neurons emerge and increase when network becomes sparse. Right: dormant neuron population increases during DST.

While DST has achieved superior performance, most DST algorithms are only focusing on algorithm level in pursue of high accuracy and theoretical computation reduction. From general optimization perspective, the effectiveness of the algorithms cannot be guaranteed due to the random and heuristic nature of the dynamic network re-routing scheme, resulting in high frequency updates on network topology to ensure the greedy search works (Evci et al., 2020). As shown in Figure 1(a), when the update frequency decreases, DST accuracy drops significantly. From system design perspective, high update frequency requires large learning rate and batch size to enforce stable neuron learningEvci et al. (2020); Liu et al. (2021c), incurring large system overhead on training devices, hence contradicting to the principles of deploying efficient training on ubiquitous devices. Here, the system overhead comes from the time spent on the computation graph reconstruction after each model typology update on the edge. Considering the model with the highest accuracy in Figure 1(a), the total of 1,200 updates translate to 114 and 686 minutes system overhead when the sparse ResNet-32 and VGG-19 models are compiled to executable code on mobile devices (Niu et al., 2020).

In accordance with the above observations, we find that the relatively low searching effectiveness is the key that causes high search frequency and any subsequent drawbacks. We identify the reason is that sparsity produces *dormant neurons* in the DST process, and such dormant neurons are extremely persistent to the "prune-grow" dynamism of DST once they are generated. These dormant neurons produce a large number of zeros, which contribute relatively little to the output of succeeding layers or to the final results (Hu et al., 2016). In Figure 1(b), we demonstrate the number of dormant neurons emerged and increased when sparsity is gradually introduced into DNN training. And Figure 1(c) shows that DST searching algorithm fails to discourage the production of dormant neuron. On the contrary, dormant neuron population increases throughout training process, degrading the expressive power of neurons and wasting the valuable active weights in the network.

In this paper, we argue that dormant neurons are those convolution filters that **not only contain negative values** in large magnitude **but also result in negative outputs** after convolution and are set to zero for post-ReLU outputs. Although these neurons have weights, they cannot be effectively updated during backpropagation. When the network is sparse, the impact of such large negative weights is amplified since they account for a larger proportion to the remaining weights (see Figure 1(b)). On the other hand, the frequent pruning operation in DST cannot *surgically remove* those harmful weights since they are heuristically pruned by small magnitude (Mocanu et al., 2018; Mostafa & Wang, 2019; Dettmers & Zettlemoyer, 2019; Lym et al., 2019; Evci et al., 2020; Yuan et al., 2021; Liu et al., 2021c;a).

To address the problem, we propose Neuron Revitalization framework for DST (**NeurRev**), which comprises an innovatively re-designed *pruning* operation that identify and surgically remove large negative weights. By doing so, NeurRev successfully reduces the number of dormant neurons as shown in Figure 1(c), which facilitate more effective learning. Please note that our proposed method not solely prune the negative weight based on their magnitudes, but also based on the post-ReLU feature map. Therefore, what we suggest is essentially a *data-driven* approach, different from earlier heuristic pruning techniques, demonstrating its effectiveness. We also demonstrate that NeurRev effectively reduces the dynamic update frequency, which enables efficient training on resource-constrained mobile devices. We also discuss the dominant practice of ReLU in a software-hardware co-design perspective, exhibiting the wide applicability of NeurRev in the efficient learning field. We summarize our contributions as follows:

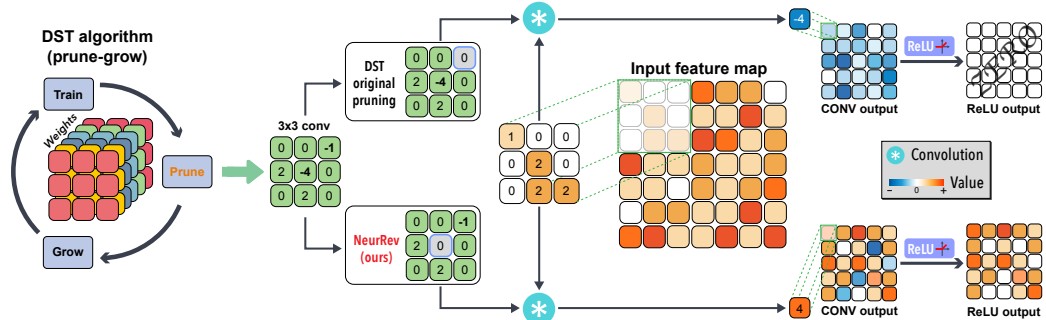

Figure 2: Overview of NeurRev. In a DST process showing in the left, NeurRev not only prunes the weight variable based on the magnitude of each weight, but also identifies and prunes the harmful weights that produce a dormant neuron (i.e., all zero output) shown on the right.

- We find dormant neurons exist in the DST training process as sparsity is introduced and increased in DNN model. We show the dormant neurons are persistent to the DST algorithms and degrading the learning effectiveness that eventually leading to impractical settings for efficient deployment.
- We develop a novel DST framework NeurRev, which identifies and surgically removes large negative weights to revitalize the dormant neurons, and achieves outperforming accuracy on CIFAR-10/100 and ImageNet compared to the state-of-the-arts.
- We deploy NeurRev on the resource-constraint mobile devices, and achieve 1.8-2.1$\times$ acceleration to dense training and a maximum of $112\times$ less overhead compared to other sparse training methods.
- We systematically evaluate ReLU activation function in a software-hardware co-design perspective, demonstrating NeurRev exhibits significant promise for extensive adoption in efficient learning field.

## 2 NEURON REVITALIZATION FOR DYNAMIC SPARSE TRAINING

In this section, We first describe the preliminary knowledge and notations of the dynamic sparse training algorithms. Then, we show that dormant neurons are produced when the neural network is sparse and put forward our algorithm to search and awake dormant neurons during sparse training. Our method is targeting practical implementation of efficient training on the edge device, therefore we also discuss NeurRev in a software-hardware co-design perspective.

### 2.1 PRELIMINARIES

Sparse training seeks to minimize the loss function $\sum_i \mathcal{L}(f(X_i; \boldsymbol{\theta}_{\mathrm{s}}), y_i)$ for a given dataset $D$ with individual samples $X_i$ and targets $y_i$. Here, $f(X_i; \boldsymbol{\theta}_{\mathrm{s}})$ is denoted as a sparse neural network. $\boldsymbol{\theta}_{\mathrm{s}}$ is a subset of a dense weight $\boldsymbol{\theta} \in \mathbb{R}^N$, where $N$ is the total number of weights in the network.

**Sparse initialization.** Current works initialize sparse neural networks mainly following two schemes: uniform (Mostafa & Wang, 2019; Dettmers & Zettlemoyer, 2019) and *Erdős-Rényi* (ER) graph (Mocanu et al., 2018; Evci et al., 2020; Liu et al., 2022b). Under the uniform setting, the sparsity of each layer is equal to the whole network sparsity. ER scheme will adaptively allocate sparsity based on the number of parameters of each layer, while ensuring the total sparsity at a certain value. Based on ER, *Erdős-Rényi-kernal* (ERK) (Evci et al., 2020) additionally takes into account the information of the convolution kernel in assigning sparsity of each layer. Other sparsity distribution such as dynamic sparse reparameterization (Mostafa & Wang, 2019) adjusts the layer-wise sparsity on-the-fly during training process. In this paper, we use the uniform and ERK sparsity since the fixed layer-wise sparsity ratio brings benefits to the hardware, and they avoid reconfiguration of the hardware resources, which indicates lower overhead and stable throughput. The overall weight sparsity can be computed with $1 - \frac{\|\boldsymbol{\theta}_{\mathrm{s}}\|_0}{\|\boldsymbol{\theta}\|_0}$, where $\|\cdot\|_0$ denotes the number of non-zero elements in a given variable.

**Training process.** During the DST process, existing methods follows a prune-grow regime to explore different network topologies (i.e., subnetworks), as shown in Figure 2 left. Suppose the total number of training iteration is $\tau_{end}$, then we conduct the DST step for the $\tau_{stop}(< \tau_{end})$ iterations at every $\Delta T_p$ iteration. DST prunes the remaining weights and then grows same number of weights back from the inactive ones to maintain sparsity ratio. Among a variety of DST algorithms, parameter growing is based on different metrics, such as gradient (Evci et al., 2020), momentum (Dettmers & Zettlemoyer, 2019), or randomness (Yuan et al., 2021). On the other hand, pruning is solely based on the magnitude of the weights. Our innovation primarily resides in the long-neglected pruning part, and random growing (Yuan et al., 2021) is adopted to reduce the peak memory consumption during the backpropagation.

## 2.2 NEURREV FOR DST – FROM RATIONALE TO ALGORITHM

### 2.2.1 WHY DORMANT NEURONS EXIST

In the case of image classification tasks, we mainly leverage ReLU-based CNNs. The weights in convolutional and fully connected layers update according to backpropagation and gradient descent algorithm in the training process. In the backpropagation process, we first compute the loss by cross entropy function. The loss $\mathcal{L}$ can be denoted as: $\mathcal{L} = -\sum_{i=1}^{n} y_i log \ \hat{y}_i$ where $y_i$ is the ground truth label of the sample, $\hat{y}_i$ is the predicted label of the model. The loss will be used in the backpropagation process to update the model weights $w_{ij}^{(l)}$ using the gradient descent (GD) algorithm. The GD algorithm can be expressed as the following equation: $w_{ij}^{(l)} = w_{ij}^{(l)} - \alpha \frac{\partial \mathcal{L}}{\partial w_{ij}^{(l)}}$ where $\frac{\partial \mathcal{L}}{\partial w_{ij}^{(l)}}$ is the gradient of $w_{ij}^{(l)}$ and can be expressed as:

$$\frac{\partial \mathcal{L}}{\partial w_{ij}^{(l)}} = \frac{\partial \mathcal{L}}{\partial z_i^{(l+1)}} \frac{\partial z_i^{(l+1)}}{\partial w_{ij}^{(l)}} = \delta_i^{(l+1)} \frac{\partial z_i^{(l+1)}}{\partial w_{ij}^{(l)}} = \delta_i^{(l+1)} a_j^{(l)}. \tag{1}$$

$w_{ij}^{(l)}$ represents the weights of neuron connections from the neuron $i$ in layer $l$ to the neuron $j$ in layer $l + 1$, and $\alpha$ stands for learning rate. $\delta_i^{(l+1)}$ means the gradient values of unit $i$ in layer $l + 1$. As shown in Figure 2, under the sparsity constrains, the negative weights among the remaining weights have more dominant effects on generating dormant neurons, which produces numerous zeros in forward propagation, and leading to $a_j^{(l)}$ in equation 1 to be zero in backpropagation. As a result, the gradient flow will be blocked by dormant neurons and the corresponding weights may not be updated.

### 2.2.2 SEARCH AND AWAKE DORMANT NEURONS

We present NeurRev, an integrated algorithm where the operations are naturally coupled with the classic DST regime. Algorithm 1 shows the outline of NeurRev, where the dormant neuron searching and awaking operations follow right after the occurrence of prune-grow step in DST, without incurring any additional update steps.

**Search dormant neurons.** Evaluating whether a neuron is a dormant neuron using layer output is computationally intensive due to the large volume of output feature maps. As we find that the gradients of dormant neurons are often zero, those neurons can hardly update weights in the training process. Based on this phenomenon, we can search for those dormant neurons according to their weight changes over a certain period of time. As shown in Algorithm 1, we use $\boldsymbol{\theta}_s^*$ to record the non-zero weights after each NeurRev step, and compute the value differences $\Delta\theta$ with the newly updated weights during the next NeurRev step. Based on $\Delta\theta$, we will sort and pick the $p \cdot 100\%$ negative weights of total weights and record their indices for the awake step. Please note that the `Prune&Grow(·)` function is the regular DST update step, but with the $s - p$ target sparsity (i.e., more active weights) for the subsequent `Search&Awake(·)` operations.

**Awake dormant neurons.** Algorithm 2 demonstrates the steps to awake the dormant neurons by pruning the harmful weights recorded by the searching step. Our empirical observation indicates that the negative weights with zero changes (i.e., $\Delta\theta = 0$) are those with relatively large magnitudes, which proves our prior analysis in Section 2.2.1. As illustrated in Algorithm 2, we awake the dormant neuron by pruning total number of $p \cdot \|\boldsymbol{\theta}\|_0$ negative weights, thus reaching the target sparsity ratio $s$. By doing so, the neurons will be released from the dormant status and regain their learning ability.

| **Algorithm 1:** NeurRev for DST | **Algorithm 2:** Search and Awake |
|---|---|
| **Input:** $\boldsymbol{\theta}_s, \boldsymbol{\theta}_s^*, s, \Delta T_p, \tau, p, \tau_{stop}$ 
 **Output:** A sparse model satisfying the target parameter sparsity $s$. 
 **Init:** Initialize $\boldsymbol{\theta}_s$ according to $s$. $\boldsymbol{\theta}_s^* = \boldsymbol{\theta}_s$. 
 **while** $\tau < \tau_{stop}$ **do** 
   **if** $\tau \ mod \ \Delta T_p == 0$ **then** 
    $\Delta \theta = \texttt{abs}(\boldsymbol{\theta}_s - \boldsymbol{\theta}_s^*)$ 
    $\boldsymbol{\theta}_s \leftarrow \texttt{Prune\&Grow}(\boldsymbol{\theta}_s, s - p)$ 
    $\texttt{Search\&Awake}(\boldsymbol{\theta}_s, \Delta\theta, p)$ 
   $\boldsymbol{\theta}_s^* = \boldsymbol{\theta}_s$ 
 Continue sparse training from $\tau_{stop}$ to $\tau_{end}$. | **Input:** $\boldsymbol{\theta}, \boldsymbol{\theta}_s, \Delta\theta, p$ 
 **Output:** A pruned sparse model. 
 **Init:** pruned_weights = 0 
 sorted_index = $\texttt{Sorted\_index}(\Delta\theta,$ ascending_order) 
 **for** $i$ **in** sorted_index **do** 
   **if** $\boldsymbol{\theta}_s[i] < 0$ **then** 
    $\texttt{Prune}(\boldsymbol{\theta}_s[i])$ 
    pruned_weights = pruned_weights + 1 
   **if** *pruned_weights* $> p \cdot \|\boldsymbol{\theta}\|_0$ **then** 
    $\texttt{exit()}$ |

## 2.3 NEURREV IN SOFTWARE-HARDWARE CO-DESIGN PERSPECTIVE

The proposed NeurRev is mainly focused on optimizing the DST process of the ReLU-based neural network. In this section, we discuss the necessity of adopting NeurRev by evaluating the performance of ReLU activation function in a software-hardware co-design perspective.

**ReLU is a versatile activation function.** ReLU activation function is pervasive in variety of network structures in computer vision tasks, such as ResNet(He et al., 2016), YOLO (Redmon et al., 2016; Redmon & Farhadi, 2017; 2018; Bochkovskiy et al., 2020), PointNet (Qi et al., 2017), Deeplab (Chen et al., 2017), etc. As other classic activation functions such as sigmoid, hyperbolic tangent (tanh), LeakyReLU (Maas et al., 2013), PReLU (He et al., 2015), etc. are also useful, ReLU is still the number one choice in the computer vision tasks due to its stable software-level performance and the ease of implementation at hardware-level. Although classic activation functions like sigmoid and tanh are well supported in many edge platforms (e.g., ARM-based processor in mobile phones), they still suffer from limited accuracy due to gradient vanishing issue caused by saturated output range (van). Besides, extra operations such as exponentiation and division are introduced (e.g., sigmoid, tanh, ELU (Clevert et al., 2016), SELU (Klambauer et al., 2017) ,etc), which potentially increases the computation overhead.

Through rigorous investigation and more sophisticated searching algorithms (Ramachandran et al., 2017), new activation functions are developed to enhance the performance and capabilities of DNNs. However, the hardware implementation of such activation functions either employs the look-up table and Taylor expansion method (Timmons & Rice, 2020), or uses the piecewise linear function method (Ngah et al., 2016; Zhengbo et al., 2020) to approximate the non-linearity of the original activation functions, which essentially a trade-off between the degree of approximation and hardware performance. For instance, HardSwish (Howard et al., 2019) replaces the computationally expensive sigmoid with a piecewise linear analogue, but it degrades the accuracy during implementation. LeakyReLU, PReLU (He et al., 2015) and RReLU (Xu et al., 2015) either experience unstable learning, or incorporate learnable parameters which bring higher implementation complexity. We present the experimental comparison of different activation functions in this software-hardware co-design perspective in Section 3.3.

**Our framework.** To support efficient and accurate training on edge devices, especially the resource-constraint mobile devices, we use the compiler code generation (Niu et al., 2020) to convert DNN computation graph into *static* code (e.g., OpenCL or C++) for direct execution. During each sparse topology update, computation graph is reconstructed and the execution code is recompiled consequently, which incurs recompilation time overhead that elongates the total training time. Undoubtedly, the less topology update occurs, the less overhead there is for the overall training time. In our framework, NeurRev achieves notably low update frequency by fixing the most significant drawback that ReLU brings, while preserving the software- and hardware-level advantages (e.g., stable, easy implementation) that ReLU comprises. By enhancing the learning ability of sparse neurons, the searching for the optimal network topology can be more effective, thus reducing the number of iterations and hardware resources. Please note that the principle of awaking the dormant neuron can also be extended to non-ReLU network and tasks.

Table 1: Test accuracy of pruned ResNet-32 on CIFAR-10/100.

| Datasets | Sparsity Distribution | CIFAR-10 | | | CIFAR-100 | | |
|---|---|---|---|---|---|---|---|
| Pruning ratio | | 90% | 95% | 98% | 90% | 95% | 98% |
| **ResNet-32** | dense | 94.88 | 94.88 | 94.88 | 74.94 | 74.94 | 74.94 |
| LT [12] | non-uniform | 92.31 | 91.06 | 88.78 | 68.99 | 65.02 | 57.37 |
| SNIP [23] | non-uniform | 92.59 | 91.01 | 87.51 | 68.89 | 65.02 | 57.37 |
| GraSP [54] | non-uniform | 92.38 | 91.39 | 88.81 | 69.24 | 66.50 | 58.43 |
| Deep-R [3] | non-uniform | 91.62 | 89.84 | 86.45 | 66.78 | 63.90 | 58.47 |
| SET [34] | non-uniform | 92.30 | 90.76 | 88.29 | 69.66 | 67.41 | 62.25 |
| DSR [35] | non-uniform | 92.97 | 91.61 | 88.46 | 69.63 | 68.20 | 61.24 |
| RigL-ITOP [29] | uniform | 93.19 | 92.08 | 89.36 | 70.46 | 68.39 | 64.16 |
| RigL [11] | uniform | 93.07 | 91.83 | 89.00 | 70.34 | 68.22 | 64.07 |
| MEST+EM [58] | uniform | 92.56 | 91.15 | 89.22 | 70.44 | 68.43 | 64.59 |
| **NeurRev (ours)** | uniform | **93.31±0.11** | **92.18±0.14** | **89.96±0.12** | **70.87±0.08** | **68.77±0.12** | **64.91±0.06** |
| RigL [11] | ERK | 93.55 | 92.39 | 90.22 | 70.62 | 68.47 | 64.14 |
| RigL-ITOP [29] | ERK | 93.70 | 92.78 | 90.40 | 71.16 | 69.38 | 66.35 |
| **NeurRev (ours)** | ERK | **93.84±0.09** | **92.93±0.14** | **90.84±0.12** | **71.96±0.06** | **69.92±0.05** | **66.82±0.07** |

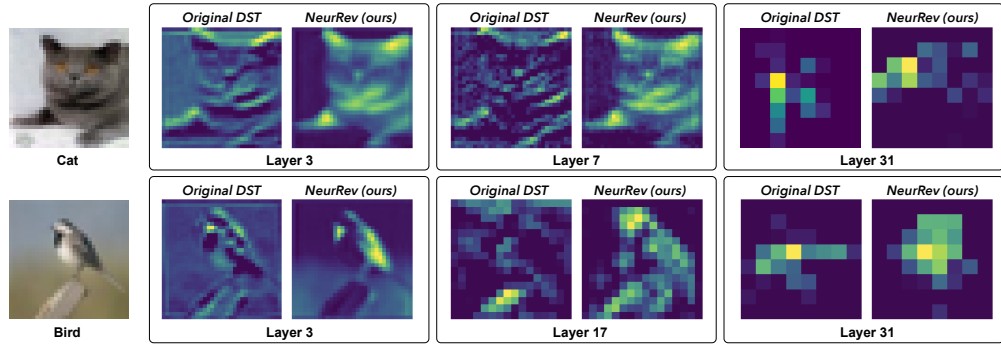

Figure 3: Neuron outputs at different layers of a ResNet-32 with 90% sparsity on CIFAR-10 images.

# 3 EXPERIMENTAL RESULTS

In this section, we carry out extensive experiments to demonstrate the advantages of NeurRev. We implement our framework based on ResNet-32 (Wang et al., 2020) and VGG-19 (Simonyan & Zisserman, 2014), training and testing on CIFAR-10/100 (Krizhevsky, 2009). In order to show that NeurRev also applies to large datasets, we train and test our framework on ImageNet (Russakovsky et al., 2015) based on ResNet-34/50 (He et al., 2016). The training speed results are obtained using a Samsung Galaxy S21 with Snapdragon 888 chipset. We repeat software training experiments for 3 times and report the mean and standard deviation of the accuracy.

**Experiment Setting** We follow the setting of GraSP (Wang et al., 2020) to set our training epochs equal to 160 for CIFAR-10/100. For ImageNet, we use both standard 100 training epochs and extended training epochs to make comprehensive comparison with other methods. We use the SGD optimizer and apply the cosine annealing learning rate schedule during training. For CIFAR-10/100, we set a batch size of 64 and the initial learning rate to 0.1. For the ImageNet software results, we set the batch size to 1024 and the learning rate to 1.024. The learning rate is scheduled to have a linear warm-up for 5 epochs before reaching the initial learning rate value. In sparsity distribution settings, we carry out experiments on both uniform and ERK distributions. For the uniformly sparse distribution, we set the first convolution layer dense and the sparsity of the remaining convolutional layers to the value of overall sparsity. For ERK distribution, we follow the setting of RigL (Evci et al., 2020). The detailed experiment setting and hyper-parameter selections can be found in Appendix A.

## 3.1 ACCURACY RESULTS

**CIFAR-10 and CIFAR-100.** The NeurRev results on ResNet-32 and VGG-19 are shown in Table 1 and Appendix B. We compare NeurRev with different sparse training methods at 90%, 95% and 98%

Table 2: Results of ResNet-50 on ImageNet-1K.

| Method | Sparsity Distribution | Top-1 Accuracy | Training FLOPs | Inference FLOPs | Top-1 Accuary | Training FLOPs | Inference FLOPs |
|---|---|---|---|---|---|---|---|
| ResNet-50 | dense | 76.9 | (×e18) | (×e9) | 76.9 | (×e18) | (×e9) |
| Sparsity | | | 80% | | | 90% | |
| LT [12] | | 72.6 | n/a | 2.7 | 70.1 | n/a | 1.7 |
| SNIP [23] | non-uniform | 69.7 | 1.67 | 2.8 | 62.0 | 0.91 | 1.9 |
| GraSP [54] | | 72.1 | 1.67 | 2.8 | 68.1 | 0.91 | 1.9 |
| Deep-R [3] | | 71.7 | n/a | n/a | 70.2 | n/a | n/a |
| SNFS [23] | non-uniform | 73.8 | n/a | n/a | 72.3 | n/a | n/a |
| SET [34] | | 72.9 | 0.74 | 1.7 | 69.6 | 0.10 | 0.1 |
| DSR [35] | | 73.3 | 1.28 | 3.3 | 71.6 | 0.96 | 2.5 |
| RigL [11] | | 75.1 | 1.34 | 3.4 | 73.0 | 0.80 | 2.0 |
| RigL$_{5\times}$ [11] | ERK | 77.1 | 6.69 | 3.4 | 76.4 | 3.94 | 2.0 |
| **NeurRev** | | **75.7±0.09** | **1.34** | 3.4 | **73.4±0.06** | **0.80** | 2.0 |
| **NeurRev$_{3\times}$** | | **77.6±0.10** | **4.01** | 3.4 | **76.7±0.09** | **2.38** | 2.0 |
| RigL [11] | | 74.6 | 0.74 | 1.7 | 72.0 | 0.39 | 0.9 |
| MEST$_{0.7\times}$ [58] | | 75.4 | 0.74 | 1.7 | 72.6 | 0.39 | 0.9 |
| Top-KAST [20] | uniform | - | - | - | 73.0 | 0.63 | 0.9 |
| SpFDE [59] | | 75.4 | 0.74 | 1.7 | - | - | - |
| **NeurRev** | | **75.9±0.17** | **0.74** | 1.7 | **73.3±0.11** | **0.39** | 0.9 |
| MEST$_{1.7\times}$ [58] | uniform | 76.7 | 1.84 | 1.7 | 75.9 | 0.80 | 0.9 |
| **NeurRev$_{1.5\times}$** | | **76.7±0.08** | **1.10** | 1.7 | **76.1±0.10** | **0.48** | 0.9 |
| RigL$_{5\times}$ [11] | uniform | 76.6 | 3.71 | 1.7 | 75.7 | 1.95 | 0.9 |
| **NeurRev$_{3\times}$** | | **77.2±0.15** | **2.28** | 1.7 | **76.0±0.12** | **0.96** | 0.9 |

sparsity. We have included the static sparse training methods such as the lottery ticket hypostasis (LTH) (Frankle & Carbin, 2019. arXiv:1803.03635), SNIP (Lee et al., 2019), and GraSP (Wang et al., 2020), and the representative DST methods (Bellec et al., 2018; Mocanu et al., 2018; Mostafa & Wang, 2019; Evci et al., 2020; Yuan et al., 2021; Liu et al., 2021c). We can notice that the accuracy of DST are generally higher than static sparse training methods, which indicates that accelerating training with sparsity should use a dynamic updating scheme. Compare with the baselines, NeurRev achieves better accuracy in both uniform and ERK sparsity distributions. We attribute the outperforming accuracy of NeurRev to the better learning ability of each sparse neuron. As shown in Figure 3, we randomly pick neurons at different layers of a ResNet-32 model. We can see that when the neuron is dormant (i.e., all or most of weights are zeros) in traditional DST, NeurRev can successfully revitalize it with meaningful output. Additionally, NeurRev can enhance the active neurons by better expressing the output, which also proves the effectiveness and importance of our proposed method.

**ImageNet-1K.** Table 2 shows the the training FLOPs and accuracy results of ResNet-50 on ImageNet-1K dataset. Compared with the SOTAs, our result shows higher test accuracy at the same computational cost level. To make fair comparison with different training recipes of RigL (Evci et al., 2020) and MEST (Yuan et al., 2021), we scale our training epochs to have the same or less overall training FLOPs for comparisons. We use the original training data to demonstrate the software-level accuracy, and the results show that NeurRev consistently outperforms other baselines in both original or scaled training recipes. Note that training an original ImageNet-1K on mobile device is not practical, we also scale the input to 128×128 and evaluate the hardware performance. The results are shown in Section 3.2. More analysis and results for ResNet-34 are shown in Appendix C.

## 3.2 SYSTEM OVERHEAD AND TRAINING ACCELERATION OF NEURREV DEPLOYMENT

**System Overhead Evaluation.** For hardware performance on the mobile phone, we follow the original training settings such as training epochs in the cited baselines, but scale the batch size to match computation resources on the hardware. We use batch size of 64 for both CIFAR and ImageNet. Please note that even we change the batch size on hardware implementation, the results can still reproduce the original accuracy with gradient accumulation. Figure 4 reports the number of updates, overhead time, and accuracy for network topology updates on ResNet-32. We can clearly see that NeurRev has 12-96× less updates/overhead, while achieving better accuracy compared to others. More results on VGG-19 are shown in Appendix D.

**End-to-End Training Acceleration with System Overhead.** Figure 5 shows the comparison results of different DST methods in terms of overall training time with regards to the system overhead. We

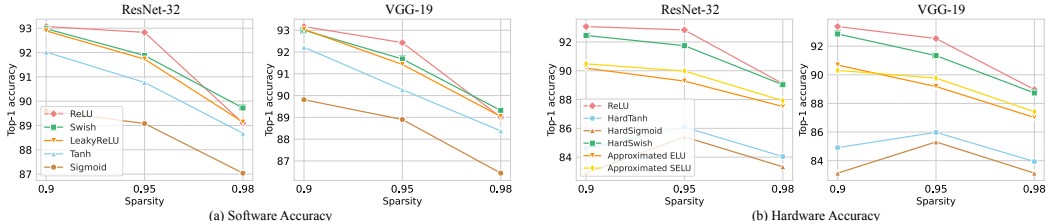

Figure 5: Training acceleration and overhead comparison with dense model and other DST methods.

Figure 6: Software and hardware evaluations on different activation functions. (a) accuracy obtained with GPU server, (b) accuracy of hardware implementation simulation.

have evaluated both mobile CPU and GPU time with CIFAR and ImageNet data. Note that original ImageNet data is not practical for edge-level training, we scale the input size to 128×128. To better demonstrate the results, we use training epochs of 100 and 30 for CIFAR and ImageNet, respectively. We include the dense model training time as one baseline, and the experimental results shows that NeurRev achieves 1.8-2.1× better acceleration rate to dense training compared with other methods. Note that the training time includes the system overhead, which in some cases, the total time in sparse training may exceed dense model training even with the present of sparsity and optimized hardware acceleration framework. Therefore, NeurRev is proved to be the one practical DST method for efficient training on edge devices.

## 3.3 EVALUATE NEURREV WITH DIFFERENT ACTIVATION FUNCTIONS

NeurRev is targeting ReLU-based DNN architectures to enhance DST training performance in both software and hardware levels. In this part, we demonstrate the advantages of using ReLU in a software-hardware co-design perspective. We replace ReLU with multiple activation functions while leaving all other settings intact. Figure 6(a) shows the software accuracy at different sparsity ratios, and ReLU has an overall better performance. When the training is performed on a mobile device as showing in Figure 6(b), most of other activation functions need to be implemented with piecewise linear approximation. Even with specifically designed hardware version (e.g., HardSwish, HardSigmoid, etc.), the accuracy on hardware still cannot match the ReLU-based DNN.

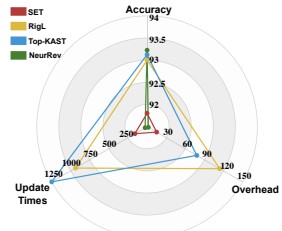

Figure 4: Overhead evaluation on ResNet-32.

## 3.4 ABLATION STUDY

▷ **Different Update Ratios With NeurRev.** We manually set the update ratio for the sparse topology searching process. The update ratio is the proportion of non-zero weights that are being pruned and grew at each update step. We set the update ratios ranging from 1% to 30% and test the accuracy. As we can see in Figure 7(a), the higher the sparsity of the network, the more sensitive it is to the update ratio. Therefore, when the network is highly sparse, the update ratio should be carefully set.

▷ **Different Update Frequencies With NeurRev.** We test NeurRev at different update frequencies in Figure 7(b). We set the update frequency from 100 iteration/update to 10 epochs/update. We can clearly see that our framework performs well at a low update frequency (e.g., 5, 10 epochs per update). Recall the results in Figure 1(a), we can conclude that NeurRev is a stable DST algorithm.

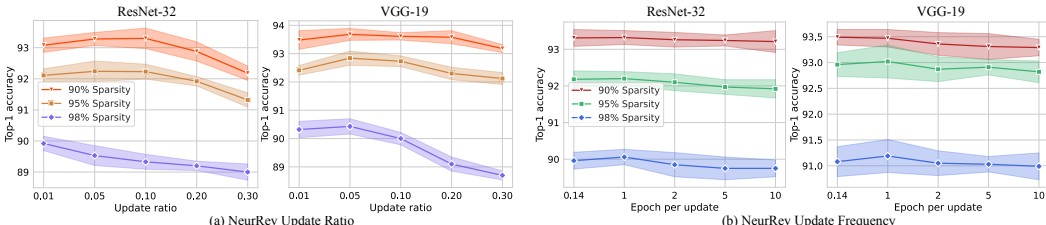

Figure 7: Different update ratios and update frequencies of NeurRev.

## 4 RELATED WORKS

Most Sparse training works can be divided into two categories, static sparse training, and dynamic sparse training. Static sparse training determines the structure of the sparse network at the initial stage of training by using the pruning algorithm. Dynamic sparse training begins with a sparse network structure that is randomly chosen at the beginning of training and alters the sparse network structure during the whole sparse training process in an effort to find a better sparse structure.

**Static Sparse Training** Static sparse training defines the structure of a sparse network through the application of a pruning algorithm . The lottery ticket hypothesis (LTH) (Frankle & Carbin, 2019. arXiv:1803.03635; Renda et al., 2020. arXiv:2003.02389; Chen et al., 2021; You et al., 2020; Frankle et al., 2020) uses iterative magnitude-based pruning (IMP) to pinpoint a subnetwork that can be trained again without losing accuracy. SNIP (Lee et al., 2019) leverages the gradients of the training loss at initialization and prunes the network at early stage of training. GraSP (Wang et al., 2020) prune connections based on the gradient flow. SynFlow (Tanaka et al., 2020) preserves the total flow of synaptic strengths which avoids layer collapse issue during pruning. FISH (Sung et al., 2021) obtains fixed subnetwork by pre-computing a sparse mask with Fisher information. Static sparse training requires full dense model forward and backward computation, which exhibits high peak memory and extensive computation on the hardware.

**Dynamic Sparse Training** Dynamic sparse training aims to minimize the computational and memory footprint during the training period. Sparse Evolutionary Training (SET) (Mocanu et al., 2018) employs magnitude-based pruning and random growth at the end of each training epoch, which is effective and easy to implement. Following this, Deep R (Bellec et al., 2018) combines dynamic sparse parameterization with stochastic parameter updates for training. However, the main result of this framework were down on small and shallow networks. Then, Dynamic Sparse Reparameterization (DSR) (Mostafa & Wang, 2019) proposes to distribute parameters between layers. Although it achieves great precision, additional training FLOPS are caused by the distributed weight that is gathered in the front layers. Sparse Networks from Scratch (SNFS) (Dettmers & Zettlemoyer, 2019) creates the sparse momentum algorithm, which finds the layers and weights that effectively reduce the error by using exponentially smoothed gradients (momentum). Same as SNFS, RigL (Evci et al., 2020) compute dense gradients when the model need to be updated to grow new connections. Top-KAST (Jayakumar et al., 2020) proposes a scalable and constant sparse DST framework for effectiveness and efficiency. Based on Top-KAST, Powerpropagation (Schwarz et al., 2021) proposes a new weight-parameterisation for neural networks, leaving the low-magnitude parameters largely unaffected by learning. ITOP (Liu et al., 2021c) investigates the underlying DST mechanism and finds that the advantages of DST result from a time-based search for all potential factors. MEST (Yuan et al., 2021) designs a memory-economic sparse training framework targeting for accurate and fast execution on edge devices. AD/AC (Peste et al., 2021) proposes a co-training of dense and sparse models method, which generates accurate sparse–dense model pairs at the end of the training process.

## 5 CONCLUSION AND DISCUSSION OF BROADER IMPACT

In this paper, we introduce a neuron revitalization framework for DST (NeurRev) to solve the long-standing dormant neuron issue that has been overlooked by most of the DST frameworks. NeurRev optimizes DST from a unique angle, which is to revitalize the dormant neuron by weight pruning, thus achieving more effective learning during DNN training. Thanks to the reduced system overhead with NeurRev, dynamic sparse training becomes practical, especially on resource-constraint edge devices. The research is scientific in nature, and we do not envision it generating any negative societal impact.

## 6 ACKNOWLEDGMENT

This work is partly supported by the National Science Foundation CCF-2312616, CCF-2008049, NASA 80NSSC23K1393, and the Army Research Office W911NF-24-1-0044. Any opinions, findings, and conclusions or recommendations expressed in this material are those of the authors and do not necessarily reflect the views of the NSF, NASA, and Army Research Office. The work of L. Guo is partially supported by NSF under grant 2008049 and 2312616. The work of L. Guo is also sponsored by the Army Research Office and was accomplished under Grant Number W911NF-24-1-0044. The views and conclusions contained in this document are those of the authors and should not be interpreted as representing the official policies, either expressed or implied, of the Army Research Office or the U.S. Government. The U.S. Government is authorized to reproduce and distribute reprints for Government purposes notwithstanding any copyright notation herein.

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

# Appendix

## A  EXPERIMENTAL SETTING

The hyperparameter settings for NeurRev are listed in Table A.1.

Table A.1: Hyperparameter settings.

| Experiments | VGG-19 on CIFAR | ResNet-32 on CIFAR | ResNet-50/34 on ImageNet |
|---|---|---|---|
| Regular hyperparameter settings | | | |
| Training epochs ($\tau_{end}$) | 160 | 160 | 100 |
| Batch size | 64 | 64 | 1024 (64 for hardware) |
| Learning rate scheduler | cosine | cosine | cosine |
| Initial learning rate | 0.1 | 0.1 | 1.024 |
| Ending learning rate | 4e-8 | 4e-8 | 0 |
| Momentum | 0.9 | 0.9 | 0.875 |
| $\ell_2$ regularization | 5e-4 | 1e-4 | 3.05e-5 |
| Warmup epochs | 5 | 0 | 5 |
| NeurRev hyperparameter settings | | | |
| Search and Awake epochs | 120 | 120 | 75 |
| Search and Awake frequency | 5 | 5 | 3 |
| Update ratio | 90% Sparsity: 10% 95% Sparsity: 5% 98% Sparsity: 5% | 90% Sparsity: 10% 95% Sparsity: 5% 98% Sparsity: 3% | 80% Sparsity: 15% 90% Sparsity: 10% |

## B  RESULTS OF VGG-19 ON CIFAR-10/100

We test NeurRev on VGG-19 using CIFAR-10 and CIFAR-100. The results are shown in Table B.1. We demonstrate results on both uniform and ERK distributions, and achieve SOTA results on CIFAR-10/100. In Figure D.1, we demonstrate more feature maps of different classes in CIFAR-10.

Table B.1: Test accuracy of pruned VGG-19 on CIFAR-10/100.

| Datasets | Sparsity Distribution | CIFAR-10 | | | CIFAR-100 | | |
|---|---|---|---|---|---|---|---|
| Pruning ratio | | 90% | 95% | 98% | 90% | 95% | 98% |
| **VGG-19** | dense | 94.20 | 94.20 | 94.20 | 74.17 | 74.17 | 74.17 |
| LT 12 | non-uniform | 93.51 | 92.92 | 92.34 | 72.78 | 71.44 | 68.95 |
| SNIP 23 | non-uniform | 93.63 | 93.43 | 92.05 | 72.84 | 71.83 | 58.46 |
| GraSP 54 | non-uniform | 93.30 | 93.43 | 92.19 | 71.95 | 71.23 | 68.90 |
| Deep-R 3 | non-uniform | 90.81 | 89.59 | 86.77 | 66.83 | 63.46 | 59.58 |
| SET 34 | non-uniform | 92.46 | 91.73 | 89.18 | 72.36 | 69.81 | 65.94 |
| DSR 35 | non-uniform | 93.75 | 93.86 | 93.13 | 72.31 | 71.98 | 70.70 |
| RigL-ITOP 29 | uniform | 93.19 | 92.08 | 89.36 | 70.46 | 68.39 | 64.16 |
| RigL 11 | uniform | 93.12 | 92.43 | 90.65 | 71.14 | 69.02 | 64.87 |
| MEST+EM 58 | uniform | 93.07 | 92.59 | 90.55 | 71.23 | 69.08 | 64.92 |
| **NeurRev (ours)** | uniform | **93.55±0.31** | **92.92±0.24** | **90.96±0.22** | **71.82±0.29** | **69.54±0.25** | **65.31 ±0.36** |
| RigL 11 | ERK | 93.77 | 92.75 | 90.87 | 71.34 | 69.21 | 65.02 |
| RigL-ITOP 29 | ERK | 93.81 | 92.81 | 90.53 | 71.46 | 69.58 | 66.72 |
| **NeurRev (ours)** | ERK | **93.88±0.29** | **93.11±0.34** | **91.17±0.22** | **72.06±0.16** | **69.95±0.25** | **67.03 ±0.27** |

## C  RESULTS OF RESNET-34 ON IMAGENET-1K

Table C.1 shows the result of NeurRev on ImageNet-1K using ResNet-34 as backbone. The results include regular training with 100 epochs, $1.5\times$ (150 epochs) and $3\times$ (300 epochs).

Table C.1: Results of ResNet-34 on ImageNet-1K.

| Method | Sparsity Distribution | Top-1 accuracy (%) | |
|---|---|---|---|
| ResNet-34 | dense | 74.10 | |
| Sparsity ratio | | 80% | 90% |
| NeurRev | uniform | 72.71 | 70.30 |
| NeurRev$_{1.5\times}$ | uniform | 73.02 | 70.56 |
| NeurRev$_{3\times}$ | uniform | 73.32 | 71.12 |
| NeurRev | ERK | 73.03 | 70.68 |
| NeurRev$_{1.5\times}$ | ERK | 73.27 | 70.92 |
| NeurRev$_{3\times}$ | ERK | 74.06 | 71.87 |

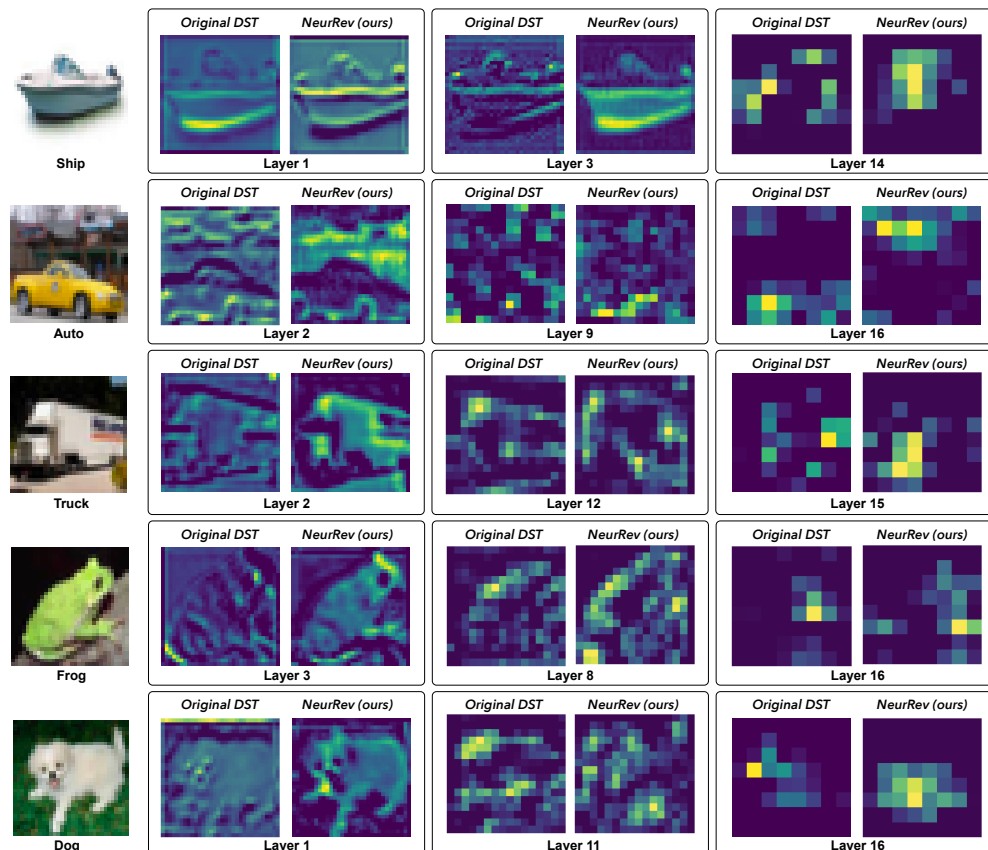

Figure D.1: Neuron outputs at different layers of a VGG-19 with 90% sparsity on CIFAR-10 images.

# D HARDWARE OVERHEAD EVALUATION RESULTS OF VGG-19

In Table D, we show the recompile overhead of some popular DST frameworks on VGG-19. The right-side radar map serves as a visual example to illustrate how our method maintains good speed with little recompilation overhead.

Table D.1: Overhead evaluation on VGG-19.

| Model | | | VGG-19 | |
|---|---|---|---|---|
| Method | Update Times | Accuracy | Recompilation Overhead (min) | |
| SET (Mocanu et al., 2018) | 160 | 92.46 | 88.2 | |
| Top-KAST (Mostafa & Wang, 2019) | 1248 | 93.08 | 686.4 | |
| RigL (Evci et al., 2020) | 936 | 93.12 | 468.1 | |
| Ours | 26 | 93.55 | 13.1 | |
| Ours | 13 | 93.47 | 6.1 | |

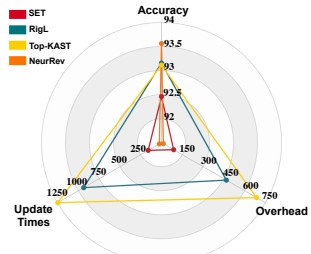

