# OpenReview forum: "NeurRev: Train Better Sparse Neural Network Practically via Neuron Revitalization"
_ICLR.cc/2024/Conference — ICLR 2024 poster_

### Official Review · Reviewer_AQcf · 2023-10-31

**Soundness:** 2 fair
**Presentation:** 3 good
**Contribution:** 2 fair
**Rating:** 6
**Confidence:** 3

**Summary:**

The paper introduces NeurRev, a novel pruning method that targets dormant neurons by assessing the change in weights during training. The authors have demonstrated the effectiveness of NeurRev through extensive software results and simulations on edge devices, showcasing its superior performance in comparison to various baselines.

**Strengths:**

1. The edge device simulations is a strength, as it demonstrates the practical applicability of NeurRev in real-world scenarios.
2. NeurRev is well-motivated and is supported by visualizations of neuron outputs. The experimental results further solidify NeurRev’s promising performance across different benchmarks.

**Weaknesses:**

My primary concern is about unfair comparison. It seems that results of Table 2 are mainly from the original papers. And I see a rarely-seen learning rate of 1.024 and an optimized cosine learning rate scheduler for the ImageNet. I am worried that the performance boost is due to the optimized training recipe. Could the authors confirm that at least Table 1 is conducted with the same training recipe?

Minor issues:
1. Page 8 Section 3.2: Figure D -> Figure 5.

I am happy to adjust my rating if my questions are addressed.

**Questions:**

1. Could the authors clarify why the weight change is zero for some weights? I ask the question sice the author mentions that weight decay is used (See appendix A).
2. For Figure 5, how are the update frequency chosen for baselines?

---

> ### Author Response · Authors · 2023-11-19
>
> Dear Reviewer AQcf,
>
> Thank you for your review and suggestions on our paper.
>
> **typo:Page 8 Section 3.2: Figure D $\rightarrow$ Figure 5.**
>
> Thank you for pointing it out. We will fix the typo.
>
> **W1: about the fair settings in comparison**
>
> Thank you for the question. Our experimental settings are carefully selected and do not incur unfair comparisons. From RigL [R1], the DST method starts to use large batch size (e.g., 4096) for better learning performance. However, in order to match such a large batch size, the learning rate should also scale up to enable effective learning with such a huge amount of information flow within each batch. According to one of the most popular (i.e., over 11.7K stars) training recipes from the NVIDIA DeepLearningExample repo on GitHub [R3], the default training batch size is set to 256 with the corresponding learning rate of 0.256, and it suggests "for other batch sizes we linearly scale the learning rate". When the batch size is scaled up or down due to different GPU resources, the learning rate is also scaled with the same scaling factor. Such training settings can be found in many published literature such as [R2][R5][R6], where batch size 2048 with a learning rate 2.048 and other settings can be found accordingly. Their results align with our own experiences that the proper scaling of learning rate and batch size will not affect final accuracy. Therefore, the performance gain does not come from the specific training recipe, as other baseline works we compare are all using a similar strategy.
>
> For Table 1 which demonstrates CIFAR-10/100 results, our training recipe inherits from GraSP [R4], and MEST also adopts GraSP's training recipe, too. For all other baseline results in Table 1, they were collected from GraSP's original paper and all use the same training recipe as GraSP does. Therefore, all comparisons in our Table 1 are fair. We will discuss the setting in our final revision.
>
> [R1] Evci U, Gale T, Menick J, et al. "Rigging the lottery: Making all tickets winners." ICML 2020.
>
> [R2] Yuan, Geng, et al. "Mest: Accurate and fast memory-economic sparse training framework on the edge." NeurIPS 2021.
>
> [R3] https://github.com/NVIDIA/DeepLearningExamples/blob/master/PyTorch/Classification/ConvNets/resnet50v1.5/README.md
>
> [R4] Wang C, Zhang G, Grosse R. "Picking winning tickets before training by preserving gradient flow." ICLR 2020.
>
> [R5] Ma, X., et al. "Effective model sparsification by scheduled grow-and-prune methods." ICLR 2022.
>
> [R6] Ma, X., Yuan, G., et al. Sanity checks for lottery tickets: Does your winning ticket really win the jackpot? NeurIPS 2021.
>
> **Q1: the weight change is zero for some weights**
>
> We are sorry for the confusion we made. We sincerely thank the reviewer for such thorough examination of our work. This is where we are not being precise enough in our writing. You are perfectly right that weight decay will cause changes on weight value, and we will revise our claim in Section 2.2 from "zero weight changes" to "close to zero weight changes". However, we want to assure the reviewer that our experimental results are 100\% correct since our code to detect dormant neurons is based on **small change** instead of zero change of weight value. Such that the conclusion based on our results still stands. We will publish our code in our final version of the paper.
>
> **Q2: update frequency chosen for baselines**
>
> Thanks for the question. For the update frequency of the baseline methods, we use the one as given in their original paper to guarantee all methods are performing under their best accuracy settings. For RigL [R1] and DSR [R2], the update frequency is every 100 iterations. These two methods require a high update frequency to maintain good mask search effectiveness (i.e., more updates have better chance to find better mask), which leads to huge end-to-end on-device training overhead due to their heuristic trial and error strategy. If we lower the update frequency, the accuracy of these methods drops dramatically. Our proposed framework NeurRev revitalizes the dormant neuron in dynamic sparse training, thus achieving more effective searching of the sparse mask and better learning ability of the sparse network. Therefore, NeurRev reduces the dynamic mask update frequency without degrading the accuracy, consequently achieving much lower overhead.
>
> [R1] Evci U, Gale T, Menick J, et al. "Rigging the lottery: Making all tickets winners." ICML 2020.
>
> [R2]Mostafa H, Wang X. "Parameter efficient training of deep convolutional neural networks by dynamic sparse reparameterization." ICML 2019.

---

> > ### Comment · Reviewer_AQcf · 2023-11-20
> > **Thank you for your response**
> >
> > Thank you for the detailed rebuttal, which has addressed most of my initial concerns. I have adjusted my rating from 5 to 6.

---

> > > ### Author Response · Authors · 2023-11-20
> > >
> > > Dear Reviewer AQcf,
> > >
> > > We extend our gratitude for raising our score. We're sincerely thankful for your support! We will include all the updates in our revision.
> > >
> > > Best regards,
> > >
> > > The Authors

---

### Official Review · Reviewer_5GHY · 2023-11-01

**Soundness:** 3 good
**Presentation:** 3 good
**Contribution:** 3 good
**Rating:** 6
**Confidence:** 4

**Summary:**

The paper presents a framework for Dynamic Sparse Training (DST), referred to as NeurRev (Neuron Revitalization). The paper addresses the dormant neuron issue in DST, a problem that has been generally overlooked in existing DST frameworks. By employing weight pruning, NeurRev revitalizes dormant neurons, leading to more effective learning during deep neural network (DNN) training. The paper also highlights the practicality of implementing NeurRev on resource-constrained edge devices due to reduced system overhead. The work is comprehensive and places itself well within the existing literature on sparse training methods, discussing both static and dynamic approaches.

**Strengths:**

1. This work identifies the interesting dormant neuron problems.
2. The authors also draw a connection between DST's update frequency and its system overhead.
3. To address the above observations, this work innovatively uses post-Relu results to prune convolution weights.

**Weaknesses:**

1. The paper would gain considerable strength from a more comprehensive set of empirical benchmarks. Specifically, the inclusion of real-world scenarios or case studies would offer a more holistic assessment of the method's efficacy and applicability. Furthermore, the method's robustness to different data distributions remains unexplored, leaving these as gaps in the experimental design.
2. The NeurRev is very limited as it seems to only work in Relu-based CNNs. Experiments only show results on the Resnet family. Would it work on more compressed networks such as Mobilenet?

**Questions:**

1. The Search and Awake process only prunes negative weights, what would happen if there are not enough non-zero negative weights to prune? How do you identify the proportion of dormant neurons to prune?
2. Other activation layers such as leaky-Relu may not set the negative input to zero but also a very small magnitude. According to the NeurRev algorithm, it should probably work the same as the original ones. Are there any limitations on the types of DNN architectures where NeurRev can be applied? Is there any plan to extend NeurRev to other types of networks? More experiments would be helpful to understand its applicability to broader cases.
3. Could you provide more details on the computational overhead introduced by the NeurRev framework? The evaluation setup is not very clear. Explain more details about the simulation.
4. What are the computational complexities involved in NeurRev in terms of both time and space? Are there any trade-offs?
5. How robust is NeurRev to different optimization algorithms? Is it specifically designed to work best with certain optimizers?

---

> ### Author Response · Authors · 2023-11-19
>
> ## **Rebuttal (1/2)**
>
> Dear Reviewer 5GHY,
>
> We sincerely appreciate your thoughtful comments about our work. The issues you have raised are very important and deserve further discussion.
>
> **W1: different data distributions**
>
> Thank you for pointing it out. Yes, we agree with your opinion that data distributions are worth more exploration. Therefore, we perform additional experiments on the Street View House Numbers (SVHN) and CelebA datasets. Specifically, CelebA is a large-scale face attribute dataset with more than 200K celebrity images. The distribution of these two datasets is different from CIFAR and ImageNet in our paper. Also, these two datasets are closer to real-world scenarios and applications. As shown in the following tables, we test our proposed NeurRev and compare it with RigL and MEST at 90\% sparsity. For CelebA, the target attribute we choose for training is Blond\_Hair.
>
> |SVHN | ResNet-32 | (90% sparsity) ||
> | :---: | :---: | :---: | :---: |
> | Dense | MEST+EM | RigL | NeurRev |
> | 95.89 | 95.32 | 95.43 | 95 . 85 |
>
> |CelebA| ResNet-18 | (90% sparsity) | |
> | :---: | :---: | :---: | :---: |
> | Dense | MEST+EM | RigL | NeurRev |
> | 95.02 | 94.18 | 94.10 | 94.98 |
>
> **W2:  Result on more compressed networks such as MobileNet**
>
> Thanks for the question. For MobileNet-V2, we train and test on CIFAR-10. The test accuracy for dense training is 94.10\% for 160 epochs. We set the sparsity at 90\% for dynamic sparse training. The test accuracy for NeurRev is 90.73\%, which is 0.56\% and 0.61\% higher than RigL and MEST+EM, respectively. The detailed results are shown in the following table.
>
> | CIFAR-10 | MobileNet-V2 | (90% sparsity) |  |
> | :---: | :---: | :---: | :---: |
> | Dense | MEST+EM | RigL | NeurRev |
> | 94.10 | 90.12 | 90.17 | 90.73 |
>
> **Q1: What would happen if there are not enough non-zero negative weights to prune? How do you identify the proportion of dormant neurons to prune?**
>
> Thank you for the good question. We investigate the sparse weight distribution in ResNet and VGG, and we find that the remaining weight distribution is very close to the Gaussian distribution for most of the cases. Therefore, the proportion of negative weight is around 50\%, and it is highly unlikely that there are not enough negative weights to be pruned since the update ratio to revitalize neurons is set to far less than 50\% of the remaining weights. We find that by setting $p$ to satisfy pruning around 10\% of the remaining weights will have better performance for sparse network training. We will add more relative discussion on this part to our final version of the paper.
>
> **Q2: Are there any limitations on the types of DNN architectures where NeurRev can be applied? Is there any plan to extend NeurRev to other types of networks?**
>
> As far as we have investigated, we did not find any limitations to the application of NeurRev. We find that many widely used activation functions inhibit the negative values in the feature map, such as ReLU, SiLU, GeLU, etc., which will inevitably cause a dormant neuron phenomenon in the dynamic sparse training settings. Our proposed NeurRev is able to detect and enhance weak activation that causes downgraded learning performance in sparse training, so it has a wide range of applicability.
>
> We extend our current approach to transformer-based network architectures. We use DeiT[R1] as the backbone for training and testing on ImageNet-1K. Our method follows the experiment setting in [R2] and compares with SViTe on DeiT-Tiny and DeiT-Small. Supported by NeurRev, we achieve 0.79\% better accuracy on DeiT-Tiny with 50\% sparsity and 1.03\% better accuracy on DeiT-Small with 70\% sparsity. Due to limited time, we did not show more models and more sparsity results. We will refine this section and add it to the paper and Appendix.
>
> | Methods | Sparsity | Accuracy |
> | :---: | :---: | :---: |
> | DeiT-Tiny | 0 \% (dense) | 72.20 |
> | SViTe [R2] | 50 \% | 70.18 |
> | NeurRev(Ours) | 50 \% | 70.97 |
>
> | Methods | Sparsity | Accuracy |
> | :---: | :---: | :---: |
> | DeiT-Small | 0 \% (dense) | 79.90 |
> | SViTe [R2] | 70 \% | 78.18 |
> | NeurRev(Ours) | 70 \% | 79.21 |
>
> [R1] Touvron H, Cord M, Douze M, et al. Training data-efficient image transformers \& distillation through attention
>
> [R2] Chen T, Cheng Y, Gan Z, et al. Chasing sparsity in vision transformers: An end-to-end exploration
>
>
>
> ## **For Q3, Q4, and Q5, please see the Rebuttal (2/2). Thanks.**

---

> ### Author Response · Authors · 2023-11-19
>
> ## **Rebuttal (2/2)**
>
> **This part follows the previous rebuttal (1/2).**
>
> **Q3:Could you provide more details on the computational overhead introduced by the NeurRev framework?**
>
> Sorry for the confusion. The computational overhead of NeurRev mainly comes from the sorting and traversing weight changes in our proposed algorithm. The time complexity is O(nlogn) (please also refer to Q4 for a detailed explanation). The time complexity for backpropagation is O($mn^2$), with $m$ stands for the number of mini-batch. For example, we roughly have more than 10,000 mini-batch for each epoch when training on ImageNet, and the update frequency for NeurRev is every 5 epochs. Therefore, the computation cost of NeurRev is less than 1/10,000 (0.01\%) of the training cost. From our analysis, we can conclude that NeurRev's search and awake scheme is very efficient, and can be considered negligible in terms of computational overhead.
>
> **Q4:What are the computational complexities involved in NeurRev in terms of both time and space? Are there any trade-offs?**
>
> Thank you for the question. The complexity of our algorithm is mainly needed to rank weight changes and In our implementation, we use the torch.sort() function [R1]. As the sort function uses quick sort, the time complexity is O(nlogn) and the space complexity is O(logn). The "trade-offs" can refer to different objectives in our paper. Since we are not quite clear about what does the reviewer refers to, we will discuss all possible "trade-offs". If the reviewer means trading off time with the space complexity of the algorithm, then our method does not have such a trade-off. If the reviewer refers to trade-offs as using more time and space to achieve better sparse training performance, then we can see from Q3 that our method has negligible computational overhead. More importantly, since we adopt such a method, the overall end-to-end on-device training experiences even lower re-compilation overhead that is required for dynamic sparse training (Section 2.3 in our paper). Therefore, our method not only avoids trading off higher time and space complexity for better performance but also improves both accuracy and efficiency for end-to-end training.
>
> [R1]https://pytorch.org/docs/stable/generated/torch.sort.html
>
> **Q5:How robust is NeurRev to different optimization algorithms? Is it specifically designed to work best with certain optimizers?**
>
> We test our method on Adam [R1] and RMSprop [R2] on CIFAR10 with ResNet-32 at 90\% sparsity. For Adam optimizer, our method achieves 1.39\% and 1.02\% higher accuracy than RigL and MEST+EM, respectively. For RMSprop optimizer, our method achieves 0.99\% and 1.06\% higher than RigL and MEST+EM, respectively. The results show that our method still works and outperforms other baseline DST methods using different optimizers.
>
> [R1] Kingma D P, Ba J. Adam: A method for stochastic optimization
>
> [R2] Hinton, G., Srivastava, N., and Swersky, K. Lecture 6d-a separate, adaptive learning rate for each connection

---

> > ### Comment · Reviewer_5GHY · 2023-11-23
> > **Responses**
> >
> > Thank you for your detailed response to my initial review comments. The additional experiments and clarifications provided in the rebuttal have been helpful in addressing some of the key issues. Here is a brief summary of my feedback:
> >
> > Data Distributions and Real-world Scenarios:
> > The additional experiments conducted on the Street View House Numbers (SVHN) and CelebA datasets are commendable. They help in demonstrating the applicability of NeurRev in more varied scenarios, enhancing the generalizability of the findings.
> >
> > Results on Compressed Networks like MobileNet:
> > The inclusion of MobileNet-V2 results adds value to the paper by showing NeurRev's effectiveness in more compressed network architectures. This addresses one of the key concerns about the framework's versatility.
> >
> > Sparse Weight Distribution Analysis:
> > Your explanation regarding the pruning strategy based on weight distributions is insightful and helpful. Providing a more in-depth statistical analysis or theoretical justification in the paper would indeed be beneficial.
> >
> > Applicability to Other Network Architectures:
> > Extending NeurRev to DeiT is a positive step, showing its broader applicability.
> >
> > Computational Overhead and Complexity:
> > The clarification on computational overhead and complexity is clear.
> >
> > Robustness to Different Optimization Algorithms:
> > It is positive to see the robustness of NeurRev with different optimizers. Further discussion on why NeurRev performs better with these optimizers and how it might interact with other less common optimization techniques would enhance the understanding of its adaptability.

---

> > > ### Comment · Reviewer_5GHY · 2023-11-23
> > > **Responses to authors**
> > >
> > > Thanks for the rebuttal. I would recommend acceptance.

---

> > > > ### Author Response · Authors · 2023-11-23
> > > >
> > > > Dear Reviewer 5GHY,
> > > >
> > > > We wanted to express our sincere gratitude for the insightful feedback you provided on our paper. We will incorporate all the updates into both the paper and the appendix. Again, thank you for your support for accepting our paper.
> > > >
> > > > Best regards,
> > > >
> > > > The Authors

---

### Official Review · Reviewer_Fgjo · 2023-11-07

**Soundness:** 4 excellent
**Presentation:** 4 excellent
**Contribution:** 3 good
**Rating:** 8
**Confidence:** 4

**Summary:**

This paper identifies the issue of "dormant neuron" in the weight-sparse training process, which hinders the performance of DST-trained models. The paper proposes a delta-based criteria to search the dormant neurons and prune them to move them out of the dormant stage, therefore helping the convergence of the sparse model. Results on multiple models and datasets show the proposed method can make DST more stable and improve the final training performance.

**Strengths:**

1. From the novelty prespective, the paper makes novel observation on the dormant neuron, and provide novel solution of delta-based pruning criteria in DST
2. From the quality prespective, the paper is technically sound. The proposed method is well motivated, and adequate experiments are performed to show the effectiveness of the proposed method
3. The paper is overal clearly written and easy to follow
4. The inclusion of runtime overhead on real hardware further improves the significance of this paper

**Weaknesses:**

1. A relavent previous work, "Deconstructing Lottery Tickets: Zeros, Signs, and the Supermask" (NeurIPS 2019) may be worth discussing. The paper explored multiple pruning criteria for LTH, including the proposed movement criteria and a similar "magnitude increase" criteria.
2. The paper focus it's discussion on CNN models, exploring models with ReLU activations and some variants. However, transformer-based models with GeLU activation is dominating SOTA architectures. It would be great to also try the proposed method on transformer model.

**Questions:**

See Weakness

---

> ### Author Response · Authors · 2023-11-19
>
> Dear Reviewer Fgjo,
>
> Thank you for your thoughtful comments about our work. Regarding the weaknesses you raised, we believe they are important points that merit further attention.
>
> **W1:A relevant previous work**
>
> Thanks for the suggestion. The "magnitude increase" criteria proposed in [R1] prunes weight with a small magnitude change.
> However, our method is fundamentally different from the above-mentioned pruning technique.
> Our work is based on the observation that sparsity produces dormant neurons in the DST process, and these dormant neurons cannot be effectively awakened by the current "prune-grow” training pattern. Therefore, we develop a novel DST framework named NeurRev, which identifies and surgically removes large negative weights with small-magnitude changes to revitalize dormant neurons. We also compare the performance of "magnitude increase" and NeurRev on CIFAR-10 (ResNet-32, 90\% sparsity). We find out our method is 1.13\% better in accuracy (92.18 \% vs. 93.31 \%). We will add more discussion about this paper in the later revision.
>
> [R1] Zhou, Hattie, et al. Deconstructing lottery tickets: Zeros, signs, and the supermask
>
>
> **W2:transformer-based models with GeLU activation**
>
> Thank you for your suggestion. For the GeLU activation function, it shares a similar property with ReLU. Both of them have a strong inhibition for negative values in the feature map. Our proposed method is able to enhance activation, so it also works for GeLU.
> For the transformer-based models with GeLU, we use DeiT [R1] as the backbone for training and testing on ImageNet-1K. Our sparse training follows the experiment setting in [R2] and compares with SViTe on DeiT-Tiny and DeiT-Small. Supported by NeurRev, we achieve 0.79\% better accuracy on DeiT-Tiny with 50\% sparsity and 1.03\% better accuracy on DeiT-Small with 70\% sparsity. Due to limited time, we did not show more models and more sparsity results. We will refine this section and add it to the Appendix.
>
> | Methods | Sparsity | Accuracy |
> | :---: | :---: | :---: |
> | DeiT-Tiny | 0 \% (dense) | 72.20 |
> | SViTe [R2] | 50 \% | 70.18 |
> | NeurRev(Ours) | 50 \% | 70.97 |
>
> | Methods | Sparsity | Accuracy |
> | :---: | :---: | :---: |
> | DeiT-Small | 0 \% (dense) | 79.90 |
> | SViTe [R2] | 70 \% | 78.18 |
> | NeurRev(Ours) | 70 \% | 79.21 |
>
> [R1] Touvron H, Cord M, Douze M, et al. Training data-efficient image transformers \& distillation through attention
>
> [R2] Chen T, Cheng Y, Gan Z, et al. Chasing sparsity in vision transformers: An end-to-end exploration

---

### Meta-Review · Area_Chair_YDkJ · 2023-12-09

**Metareview:**

The paper proposes a method to address the dormant neurons issue in the dynamic sparse training (DST) by first identifying them and then pruning the large negative weights that lead to dormant neurons when composed with relu nonlinearities. This leads to reviving the dormant neurons during training. All reviewers agree that the paper addresses a rather unexplored issue in the DST literature. Authors have added new experiments during the rebuttal phase that have addressed reviewers' comments.

**Justification For Why Not Higher Score:**

While all reviewers are positive about the paper, their support for the paper isn't enthusiastic enough for a spotlight acceptance. While the method results in reduced training overhead, the final accuracy at a given inference cost seems to be only marginally better than the existing methods.

**Justification For Why Not Lower Score:**

All reviewers agree that the paper addresses a relevant problem of dormant neurons in sparse training.

---

### Decision · Program_Chairs · 2024-01-16

Accept (poster)